# Cell type-specific contributions to a persistent aggressive internal state in female *Drosophila*

Hui Chiu[1]*[†‡], Alice A Robie[2], Kristin Branson[2], Tanvi Vippa[2], Samantha Epstein[2], Gerald M Rubin[2]*, David J Anderson[1,3]*, Catherine E Schretter[2]*[†]

[1]Division of Biology and Biological Engineering, Tianqiao and Chrissy Chen Institute for Neuroscience, California Institute of Technology, Pasadena, United States; [2]Janelia Research Campus, Howard Hughes Medical Institute, Ashburn, United States; [3]Howard Hughes Medical Institute, California Institute of Technology, Pasadena, United States

*For correspondence:
hui.chiu@yale.edu (HC);
rubing@janelia.hhmi.org (GMR);
wuwei@caltech.edu (DJA);
schretterc@janelia.hhmi.org (CES)

[†]These authors contributed equally to this work

Present address: [‡]Department of Immunology, Yale University School of Medicine, New Haven, United States

Competing interest: The authors declare that no competing interests exist.

## eLife assessment

This study by Chiu and colleagues is a **valuable** contribution to the study of the circuitry of aggressive behaviors and of mechanisms that generate persistent behavioral states. The authors find that activation of two interconnected sets of neurons results in an increase in female aggression. The data ruling out recurrent connectivity between these clusters underlying this persistent state are **convincing**.

**Abstract** Persistent internal states are important for maintaining survival-promoting behaviors, such as aggression. In female *Drosophila melanogaster*, we have previously shown that individually activating either aIPg or pC1d cell types can induce aggression. Here we investigate further the individual roles of these cholinergic, sexually dimorphic cell types, and the reciprocal connections between them, in generating a persistent aggressive internal state. We find that a brief 30-second optogenetic stimulation of aIPg neurons was sufficient to promote an aggressive internal state lasting at least 10 minutes, whereas similar stimulation of pC1d neurons did not. While we previously showed that stimulation of pC1e alone does not evoke aggression, persistent behavior could be promoted through simultaneous stimulation of pC1d and pC1e, suggesting an unexpected synergy of these cell types in establishing a persistent aggressive state. Neither aIPg nor pC1d show persistent neuronal activity themselves, implying that the persistent internal state is maintained by other mechanisms. Moreover, inactivation of pC1d did not significantly reduce aIPg-evoked persistent aggression, arguing that the aggressive state did not depend on pC1d-aIPg recurrent connectivity. Our results suggest the need for alternative models to explain persistent female aggression.

## Introduction

Persistence is an evolutionarily conserved feature of internal states, which governs the duration of an animal's behavior beyond an inciting stimulus (*Anderson and Adolphs, 2014*; *Flavell et al., 2022*). For example, a glance at a moving muleta is sufficient to keep a Spanish fighting bull aggressive for tens of minutes. Recent studies in mammals and invertebrates have begun to uncover potential mechanisms for maintaining persistent behavioral states on various timescales. These mechanisms can be 'electronic', such as persistent firing in certain cell types or circuits, 'biochemical', such as slow

**Figure 1.** Proposed model for persistent aggressive behavior in females. Recurrent connectivity between pC1d, pC1e, and aIPg (**A**) has been proposed to generate a persistent aggressive internal state by prolonging the neural activity of pC1d or aIPg neurons (**B**) (*Deutsch et al., 2020*; *Schretter et al., 2020*). For the connectivity diagram (**A**), synapse number is noted on each arrow and no thresholds were applied between types.

The online version of this article includes the following figure supplement(s) for figure 1:

**Figure supplement 1.** Comparison of the experimental designs used in *Schretter et al., 2020* and *Deutsch et al., 2020*.

**Figure supplement 2.** Additional circuit components.

decay of second messengers or their effectors influencing neural excitability, or 'systemic', such as persistent elevation of circulating hormones or neuromodulators, and may span timescales of hundreds of milliseconds to tens of seconds to days (*Dylla et al., 2013*; *Thornquist et al., 2020*; *Thornquist et al., 2021*; *Deutsch et al., 2020*; *Flavell et al., 2013*; *Jung et al., 2020*; *Kennedy et al., 2020*; *Marques et al., 2020*; *Rhoades et al., 2019*; *Robson and Li, 2022*; *Zhang et al., 2021*). In studies of working memory in mice, persistent neural activity generated by a recurrent excitatory network has been proposed to sustain an animal's delayed response to a transient stimulation (*Guo et al., 2017*; *Inagaki et al., 2019*; *McCormick, 2001*; *Wang, 2001*). However, circuit mechanisms that generate a persistent aggressive state remain unresolved (*Guthman and Falkner, 2022*). *Drosophila melanogaster* has been a useful model for discovering cell types that control aggression (reviewed in *Hoopfer, 2016*; *Kravitz and Fernandez de la, 2015*). Whether recurrent connectivity generates persistent neural activity that sustains prolonged aggression (*Figure 1*) is now a testable hypothesis due to recent advances in fly connectomics (*Hoopfer et al., 2015*; *Scheffer et al., 2020*).

In *Drosophila* males, most aggression-promoting neurons described thus far are male-specific, limiting the applicability of the available female fly connectome for addressing this question (*Asahina, 2017*). However, recent progress in identifying cell types that control female aggression has opened up this possibility (*Deutsch et al., 2020*; *Chiu et al., 2021*; *Palavicino-Maggio et al., 2019*; *Schretter et al., 2020*). Through behavioral screens and connectomic mapping, two groups independently showed that combined optogenetic activation of *doublesex* (*dsx*)-expressing pC1d and pC1e neurons promotes aggression in females (*Deutsch et al., 2020*; *Schretter et al., 2020*). Upon co-stimulation of pC1d and pC1e for either 2 or 5 minutes, *Deutsch et al., 2020* observed persistent aggression and prolonged neuronal activity in subsets of *fruitless* (*fru*)- and *dsx*-expressing neurons (*Figure 1—figure supplement 1*). As these experiments were performed using a GAL4 line labeling both pC1d and pC1e neurons, the individual roles of pC1d or pC1e in the maintenance of persistent aggression could not be determined.

In an independent study, *Schretter et al., 2020* stimulated GAL4 lines that separately labeled pC1d and pC1e neurons, as well as those that co-labeled both pC1d and pC1e, to test their effects on female aggression. They observed that activation of pC1d neurons alone generated only time-locked aggression, while stimulation of pC1e had no effect on aggression at all (*Figure 1—figure*

*supplements 1 and 2*). *Schretter et al., 2020* also identified an additional female aggression-promoting cell type, aIPg, not directly tested in *Deutsch et al., 2020*. They observed that a 30-second stimulation of aIPg neurons, like that of pC1d+e neurons, promoted aggression that outlasted the stimulation period. However, as individual flies were not separated during the stimulation period, this study could not definitively conclude that stimulation of aIPg neurons generated a persistent aggressive internal state (*Schretter et al., 2020*).

These two studies, therefore, raised a number of questions about the mechanisms generating persistent aggressive internal states in female flies. For example, can stimulation of pC1d alone generate a persistent internal aggressive state, or is pC1e needed in combination with pC1d to see this long-lasting effect? Does the duration of the persistent state depend on the type of neuron stimulated (aIPg vs. pC1d+e), the duration of stimulation, or on other factors? It is difficult to answer these questions by combining the results of *Deutsch et al., 2020* and *Schretter et al., 2020* due to experimental differences between the two studies. For example, *Deutsch et al., 2020* used males as targets of female aggressors while *Schretter et al., 2020* used females as targets, and the two studies used different optogenetic stimulation conditions (partly summarized in *Figure 1—figure supplement 1*). In this report, we evaluate the contribution of individual cell types, specifically aIPg, pC1d, and pC1e, to generating a persistent aggressive state, under directly comparable stimulation and behavioral assay conditions, and begin to test predictions of connectome-based models for circuit mechanisms underlying persistence.

## Results
### Stimulation of aIPg neurons but not pC1d neurons promotes a persistent aggressive state in females

We used a well-established behavioral test for evaluating persistent aggressive internal states (*Deutsch et al., 2020*; *Hoopfer et al., 2015*), in which same-genotype flies expressing the red-shifted channelrhodopsin, CsChrimson (CsChR), in the specified neuronal population were separated by a sliding metal door (*Figure 2A*). Therefore, their social interactions were withheld until the doors were removed at the desired time following neuronal stimulation. This design minimizes the influence of social feedback on internal state and thereby allows us to directly assess specific neuronal stimulation effects on the duration of a fly-autonomous aggressive internal state.

A 30-second 5 Hz 655 nm stimulation was applied to activate distinct neuronal populations: aIPg, pC1d, or pC1d+e (pC1d and pC1e together; the pC1d+e driver is designated as pC1-A in *Deutsch et al., 2020*). The driver lines marking these specific cell types were previously characterized in *Schretter et al., 2020* and show similar levels of expression. Moreover, in all cases, the strength of the optogenetic stimulus used to test for persistence was similar to that needed to induce aggression during stimulation. Then, 10 minutes following the stimulation, the doors were removed to allow social interactions (*Figure 2A*). Preliminary experiments using 1-, 5-, 10-, and 30-minute delays showed persistence at 1, 5, and 10 minutes, but not at 30 minutes; therefore, we performed more detailed experiments at the 10-minute time point. Aggressive behaviors such as head butting and shoving were analyzed using the Caltech FlyTracker and a JAABA classifier (*Table 1*). We observed elevated female aggression that was detectable after 10 minutes when aIPg or pC1d+e, but not pC1d neurons, were stimulated, suggesting that the aIPg neurons alone are sufficient to promote a persistent aggressive internal state. In contrast, pC1d depends on co-stimulation with pC1e to generate a persistent effect (*Figure 2B and C*). Because stimulation of pC1e neurons alone does not produce female aggression (*Schretter et al., 2020*), its contribution to the regulation of the persistent aggression was unexpected. Collectively, these results demonstrate the sufficiency of either aIPg or pC1d+e neurons to promote a persistent aggressive state in females.

Using a different experimental paradigm, we next investigated whether social feedback from a less aggressive (group housed, as shown in *Chiu et al., 2021*) wild-type target fly can contribute to the duration of persistent aggression. In this 'mixed-pair' assay, CsChrimson-expressing females were paired with wild-type female targets in chambers without a divider, allowing for social interactions at any time (*Figure 2—figure supplement 1*). Under these conditions, attacks by the tester fly persisted following the stimulation of aIPg or pC1d+e, but not of pC1d. Activation of pC1e alone did not significantly increase aggression during or following stimulation (data not shown), as previously reported

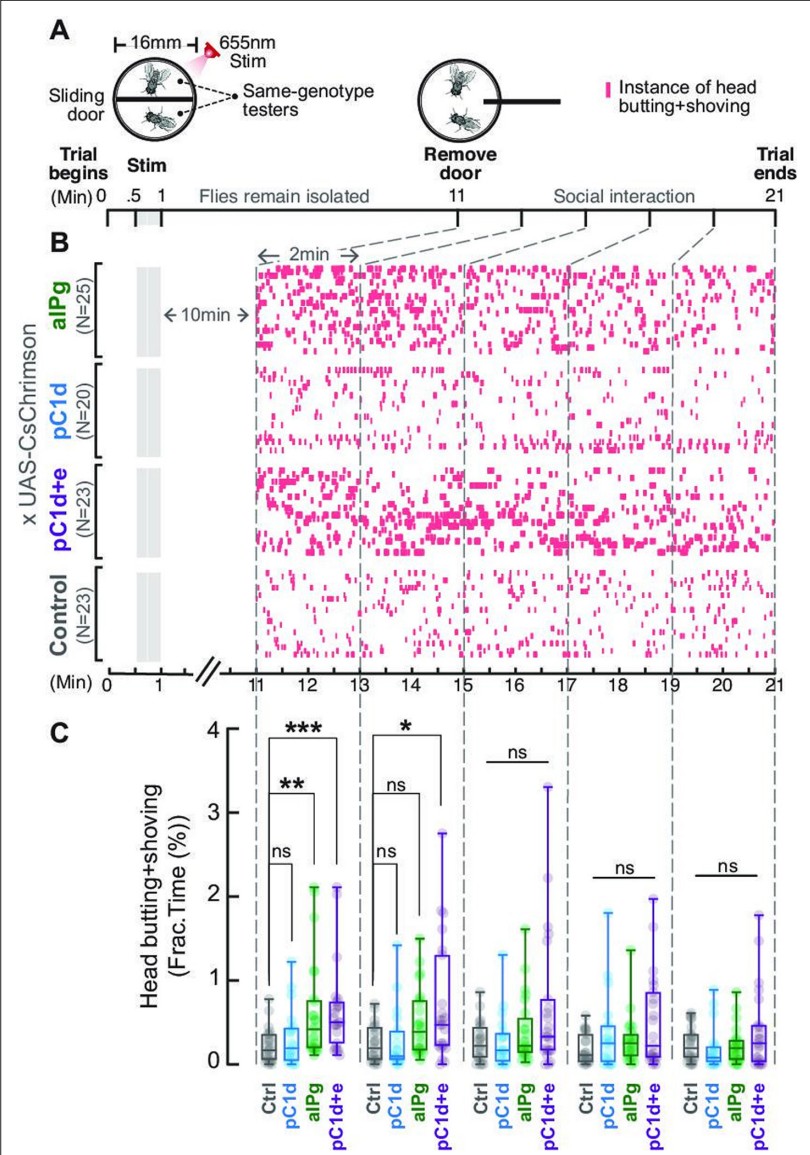

**Figure 2.** Brief stimulation of aIPg, but not pC1d alone, promotes persistent female aggression. Cell-specific contributions of aIPg, pC1d, and pC1d+e neurons to persistent aggressive internal state were tested using the sliding-door assay (**A**). Same-genotype testers were separated by a sliding door in the behavioral chamber while receiving a brief 655 nm light stimulation (13+15 seconds). Doors were removed 10 minutes after the stimulation to allow the flies to interact. Level of female aggression (head butting + shoving, red ticks) is shown in the raster plot (**B**). Fraction of time spent on fighting by each genotype female is compared in the box plot (**C**). Data were combined from three independent biological repeats. Additional information on genotypes used and statistics performed is provided in ***Supplementary files 1 and 2***. \*\*\*p<0.001; \*\*p<0.01; \*p<0.05; ns, not significant. Control, empty split-Gal4 driver (BDP-AD; BDP-DBD).

The online version of this article includes the following source data and figure supplement(s) for figure 2:

**Source data 1.** Source data for ***Figure 2*** graph on time spent performing aggressive behaviors.

**Figure supplement 1.** Stimulation of aIPg, but not pC1d alone, promotes persistent attacks toward a wild-type target.

**Figure supplement 1—source data 1.** Source data for ***Figure 2—figure supplement 1*** graph on time spent performing aggressive behaviors.

**Figure supplement 2.** Density influences the persistence of aggression.

**Figure supplement 2—source data 1.** Source data for ***Figure 2—figure supplement 2*** on percent of flies engaging in aggressive behaviors.

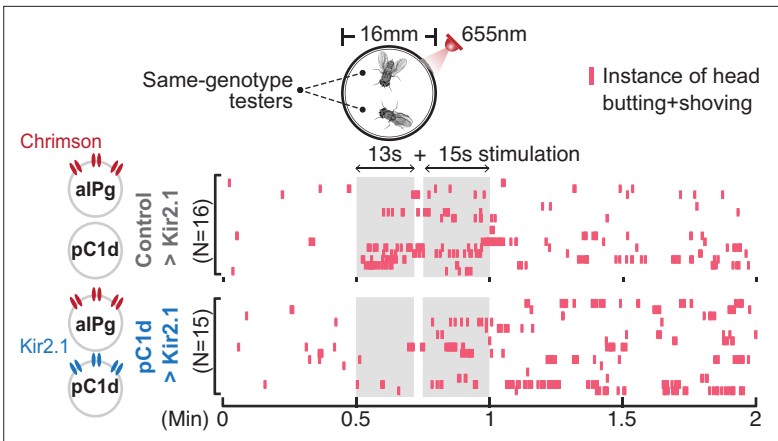

**Figure 3.** pC1d-aIPg recurrent connectivity is not required for aIPg-induced persistent aggressive behavior. Inhibition of pC1d by Kir2.1 expression did not suppress aIPg-induced persistent aggression. Aggression (head butting + shoving, red ticks) is shown in the raster plot. Flies were freely interacting throughout the experiments, and data were combined from two independent biological repeats. The differences between Control>Kir2.1 and pC1d>Kir2.1 groups in the separate 13-second and 15-second stimulation periods were not significant (13-second period, p=0.2978; 15 seconds, p=0.6650). Additional information on genotypes used is provided in *Supplementary file 1* and see *Figure 3—figure supplement 1* for quantification. Control, empty split-Gal4 driver (BDP-AD; BDP-DBD).

The online version of this article includes the following source data and figure supplement(s) for figure 3:

**Figure supplement 1.** Inhibition of pC1d activity does not alter aIPg-induced aggressive behavior.

**Figure supplement 1—source data 1.** Source data for *Figure 3—figure supplement 1* graph on time spent performing aggressive behaviors.

(*Schretter et al., 2020*). The aIPg-induced persistent aggression lasted minutes in this paradigm, which is longer than the tens of seconds-long persistence we reported previously (*Schretter et al., 2020*). As there is a sevenfold difference in the fly density per area in the two experiments (current study: 0.2 flies per sq. cm; *Schretter et al., 2020*: 0.03 flies per sq. cm) and since density is known to alter social behaviors (*Rooke et al., 2020*), we asked if the duration of persistence was influenced by the fly density in the chamber. Indeed, we found that even a modest increase in fly density (0.08 flies per sq. cm) extended persistent aggression in both aIPg and pC1d+e activated flies. These observations suggest that the frequency of social encounters can be an external factor that increases the duration of aggression (*Figure 2—figure supplement 2*).

## pC1d-aIPg functional connectivity is not required for aIPg-induced persistent aggression

We investigated whether the reciprocal pC1d-aIPg connectivity (*Figure 1A*) is required for aIPg-induced persistent aggression by performing behavioral epistasis and in vivo calcium imaging experiments. Using the expression of the inwardly rectifying potassium channel Kir2.1 (*Baines et al., 2001*), we inhibited the activity of pC1d neurons while aIPg neurons were activated for 30 seconds in freely interacting females (*Figure 3*, *Figure 3—figure supplement 1*). As aIPg-induced persistent aggression was found in the same genotype (*Schretter et al., 2020*) and mixed-pair assays (*Figure 2—figure supplement 1*), we used the same genotype pairs to investigate if pC1d inactivation altered aIPg-induced behavior. Surprisingly, chronic inhibition of pC1d neurons did not suppress aggression after aIPg stimulation, suggesting that feedback from pC1d neurons is not necessary for the persistent aggressive state promoted by aIPg neurons (*Figure 3*, *Figure 3—figure supplement 1*).

We next turned to in vivo imaging experiments to examine whether pC1d or aIPg neurons themselves exhibit persistent activity, and if so whether the strong recurrent connections between these cholinergic cells (*Scheffer et al., 2020*) contribute to such persistence. To evaluate the role of these connections, we applied either a short (5 Hz, 30 seconds) stimulus, as used in our current behavioral assays, or a long (50 Hz, 300 seconds) stimulus (as used in *Deutsch et al., 2020*) to activate aIPg or

pC1d neurons while simultaneously imaging calcium transients in either cell type using GCaMP, in head-fixed females under a two-photon microscope (*Figure 4*, *Figure 4—figure supplement 1*).

We found that neither short (30 seconds) (*Figure 4C and D*) nor long (300 seconds) (*Figure 4—figure supplement 1C and D*) stimulation of aIPg neurons triggered persistent activity in pC1d neurons or vice versa. Because aIPg neurons form reciprocal connections amongst themselves (*Figure 1A*), we also examined whether direct stimulation of aIPg neurons could cause persistent activity in these cells. We found that the response of aIPg neurons was time-locked to the stimulation period regardless of stimulation length (*Figure 4B*, *Figure 4—figure supplement 1B*). Given that a 30-second stimulation of aIPg neurons in intact flies is sufficient to elicit a persistent aggressive state that lasts longer than 10 minutes, it seems that neither aIPg nor pC1d neurons exhibit persistent activity on the time scale necessary to sustain persistent aggressive behavior. Together, our behavioral epistasis and in vivo imaging data indicate that reciprocal connections among aIPg neurons or between pC1d and aIPg neurons are neither required nor sufficient to generate a persistent internal state of aggressiveness. Therefore, persistent aggressive behavior is likely regulated by other downstream neurons that transform the output of aIPg or pC1d+e neurons.

## Concluding remarks

This work demonstrates that optogenetic stimulation of aIPg neurons at the same levels needed to produce aggression during photo-stimulation also promoted a minutes-long behaviorally latent, persistent internal state of aggressiveness (or social arousal). In contrast, pC1d neuronal stimulation with the same optogenetic parameters generated time-locked aggression but not a latent, persistent internal state. However, such a persistent internal state was observed when pC1d and pC1e were co-stimulated. Our epistasis experiments do not support a model in which persistent aggression in females is driven by pC1d neurons acting through recurrent connections with aIPg neurons. Our imaging data further indicate that neither aIPg nor pC1d neurons themselves exhibit persistent neural activity when directly or indirectly stimulated, raising the question of how the persistent aggressive state is sustained by the downstream targets of these neurons and what role pC1e plays.

Our connectomic analysis revealed that aIPg, pC1d, and pC1e neurons have both unique and shared downstream targets (*Figure 1—figure supplement 2*). pC1d and pC1e neurons form a relatively small number of synapses with one another (*Figure 1A*, *Figure 1—figure supplement 2A*); however, aIPg neurons form strong reciprocal connections with both pC1d and pC1e as well as among themselves. While aIPg neurons' recurrent synapses within the neuronal subset was a candidate for maintaining persistence, the GCaMP signals in aIPg were time-locked to the stimulus when aIPg neurons were activated. Additionally, pC1d stimulation triggered only time-locked responses in aIPg neurons (*Figure 4*).

The results reported here confirm the observation of *Deutsch et al., 2020* that co-stimulation of pC1d and pC1e promotes persistent aggression, but reveal paradoxically that stimulation of pC1d alone promotes only time-locked, but not persistent aggression. Given that stimulation of aIPg alone can promote persistent aggression in the absence of pC1d activity, these results provide evidence of distinct classes of neurons controlling aggression in a time-locked versus persistent manner. The biological significance of this distinction remains to be determined. Additionally, our data reveal a key role for pC1e in maintaining persistent aggression, despite the fact that these neurons do not produce aggression when optogenetically stimulated alone. This observation suggests that persistent aggression promoted by pC1d+e is an emergent property of their co-activation. We note that pC1e receives distinct inputs that possibly indicate the presence of another fly (*Figure 1—figure supplement 2*). While pC1e also forms recurrent connections with aIPg, these synapses constitute a relatively low proportion of the total output of aIPg (0.177%). Interestingly, the function of pC1e neurons resembles that of pCd neurons in males, in that both classes of neurons regulate the persistence of internal states but do not directly evoke aggressive behavior when stimulated (*Jung et al., 2020*). However, pCd neurons are a cell type distinct from pC1d and pC1e neurons.

This study focused on the cell type-specific contribution of aIPg, pC1d, and pC1e neurons as well as the role of their recurrent connectivity. The persistence of aggressive states could be regulated by additional circuit components not present in our driver lines or other mechanisms, including neuropeptide modulation or prior experience (*Guthman and Falkner, 2022*; *Kennedy, 2022*; *Major and Tank, 2004*). Identifying such components and their mechanisms of action will be an important

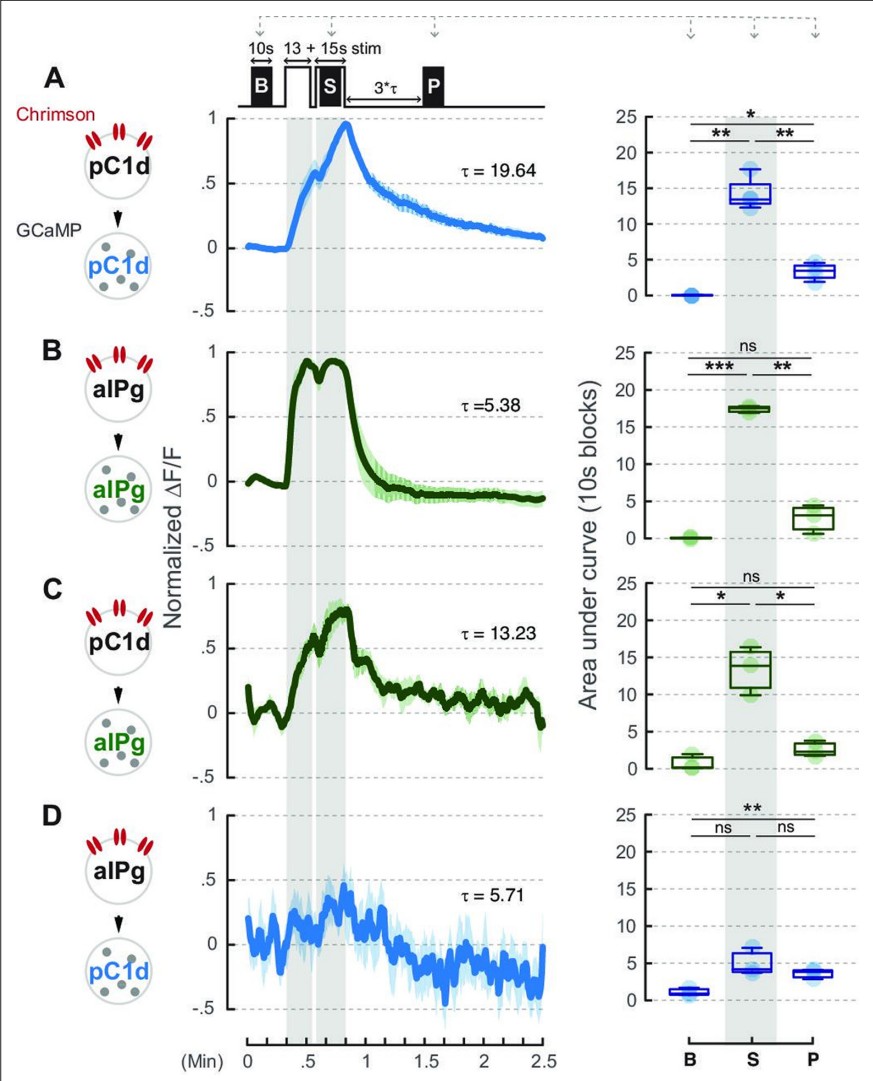

**Figure 4.** Neural activity of aIPg neurons is not prolonged by pC1d-aIPg or aIPg-aIPg recurrent connectivity. A brief 660 nm stimulation that mimics the one used in the behavioral experiment was applied to activate the Chrimson-expressing neurons: pC1d (**A** or **C**) and aIPg (**B** or **D**). GCaMP responses of pC1d (**A** or **D**) or aIPg (**B** or **C**) neurons are shown by the blue and green traces, respectively. Dark blue/green line, mean. Light blue/green line, SEM. Tau of GCaMP signals in pC1d neurons is 19.64 seconds and 5.71 seconds when pC1d or aIPg neurons were stimulated, respectively. Tau of GCaMP signals in aIPg neurons is 5.38 seconds and 13.23 seconds when aIPg or pC1d neurons were stimulated, respectively. Area under the GCaMP trace during the 10-second blocks in the baseline (B), stimulation (S), and post-stimulation (P; measured at three times Tau after stimulation) period was compared in the box plots shown at right of panel. The difference between B and P in (**D**) was not significant after multiple comparisons correction. Data were combined from three to four independent biological repeats. Additional information on genotypes used and statistics performed is provided in ***Supplementary files 1 and 2***. Light blue/green circle, individual data. ***p<0.001; **p<0.01; *p<0.05; ns, not significant.

The online version of this article includes the following source data and figure supplement(s) for figure 4:

**Source data 1.** Source data for the area under the GCaMP trace during the 10-second block periods described in *Figure 4*.

**Figure supplement 1.** A 5-minute stimulation of pC1d neurons does not induce prolonged activity in aIPg neurons.

**Figure supplement 1—source data 1.** Source data for the area under the GCaMP trace during the 10-second block periods described in *Figure 4—figure supplement 1*.

focus for future work. RNA profiling data indicate that aIPg neurons produce the neuropeptide sNPF (**Schretter et al., 2020**) and downstream targets might produce additional neuropeptides. Neuropeptides can act over the timescales needed to generate persistent aggressive behavior (**Flavell et al., 2022**). Additional mechanisms include attractor dynamics as suggested by a recent study in mice (**Nair et al., 2023**). How neural circuits implement a persistent aggressive state, and whether individual circuit elements control distinct features of the aggressive state, remain major open questions that can be addressed by future studies of this circuit.

# Materials and methods

### Key resources table

| Reagent type (species) or resource | Designation | Source or reference | Identifiers | Additional information |
|---|---|---|---|---|
| Genetic reagent (*D. melanogaster*) | w1118;;20XUAS-CsChrimson-mVenus (attP2) | *Klapoetke et al., 2014*; *Aso et al., 2014* | UAS-CsChrimson | BDSC: 55136 |
| Genetic reagent (*D. melanogaster*) | w1118;; 20xUAS-Chrimson::tdT3.1(vk5) | *Chiu et al., 2021*; Rubin lab | UAS-Chrimson | |
| Genetic reagent (*D. melanogaster*) | w1118; 20xUAS-Chrimson::tdT3.1(su(Hw)attp5) | *Watanabe et al., 2017*; Rubin lab | UAS-Chrimson | |
| Genetic reagent (*D. melanogaster*) | w1118; 13xLexAop2-CsChrimson::tdT3.1(su(Hw)attp5) | *Chiu et al., 2021*; Rubin lab | LexAop-Chrimson | |
| Genetic reagent (*D. melanogaster*) | w1118;; 13xLexAop2-CsChrimson::TdT3.1(vk5) | *Chiu et al., 2021*; Rubin lab | LexAop-Chrimson | |
| Genetic reagent (*D. melanogaster*) | w1118;; 10xUAS-eGFP::Kir2.1(attP2) | *Chiu et al., 2021*; Rubin lab | UAS-Kir | |
| Genetic reagent (*D. melanogaster*) | w1118; 20xUAS-opGCaMP6s(su(Hw)attp5) | *Chen et al., 2013* | UAS-GCaMP6s | |
| Genetic reagent (*D. melanogaster*) | w1118;; 13xLexAop2-jGCaMP7b(vk5) | *Dana et al., 2019* | LexAop-GCaMP7b | |
| Genetic reagent (*D. melanogaster*) | w1118; BPp65ADZp (attP40); BPZpGAL4DBD (attP2) | *Hampel et al., 2017* | Control (Empty-SS) | |
| Genetic reagent (*D. melanogaster*) | w1118; VT064565-p65ADZp (attP40); VT043699-ZpGAL4DBD (attP2) | *Schretter et al., 2020* | aIPg (SS36564) | Available via https://janelia.org/split-GAL4. |
| Genetic reagent (*D. melanogaster*) | w1118; 35C10-p65ADZp (JK73A), 71A09-ZpGdbd (attP2) | *Schretter et al., 2020* | pC1d (SS56987) | Available via https://janelia.org/split-GAL4. |
| Genetic reagent (*D. melanogaster*) | w1118; VT25602-p65ADZp(attP40); VT002064-ZpGal4DBD(attP2) | *Schretter et al., 2020* | pC1d+e(SS43274) | Available via https://janelia.org/split-GAL4. |
| Genetic reagent (*D. melanogaster*) | w1118; 72C11-LexA (attP40) | *Jenett et al., 2012* | aIPg-LexA | Available via https://flweb.janelia.org. |

### Fly strains

See *Supplementary file 1* for full genotypes of flies used in each figure and sources of flies. Briefly, the cell type-specific drivers were generated at Janelia Research Campus and the split-GAL4 drivers were previously reported in *Schretter et al., 2020*. Depending on the experiments, the aIPg neurons were labeled by aIPg-SS (SS36565; VT064565-p65ADZp(attP40); VT043699-ZpGDBD(attP2)) or 72C11-LexA (attp-40). Drivers for the pC1d or pC1d+e neurons were pC1d-SS (SS56987; 35C10-p65ADZp(JK73A); 71A09-ZpGDBD(attP2)) and pC1d+e-SS (SS43274; VT025602-p65ADZp (attP40); VT002064-ZpGDBD (attP2)/TM6B), respectively. Canton-S was used as wild-type target flies in mixed-pair experiments. 72C11LexA (attp40), 13xLexAop2-CsChrimson::tdT3.1 (su(Hw)attp5); 10xUAS-eGFP::Kir2.1(attp2) was generated in the Anderson lab.

### Rearing conditions

Stocks and crosses were reared at 25°C and 50% humidity and maintained on a 12 hours:12 hours light:dark cycle. Fly density was kept consistent across experiments by crossing 10–12 virgin females with 5–6 males and flipping every 2 days. Experimental flies were collected as virgins and group housed (~20 flies per vial) on vials containing retinal food (0.2 mM) in the dark. Flies were flipped into fresh vials containing food 1 day before behavioral testing.

## Behavioral assays

Unless otherwise stated, the behavioral chambers used were 6-mm-high 16-mm-diameter acrylic cylinders with a clear top and floor. The wall and lids were coated with Insect-A-Slip and silicon fluid, respectively. The floor was covered with freshly prepared apple juice agar (2.5% [w/v] sucrose and 2.25% [w/v] agarose in apple juice) and illuminated with an 850 nm backlight (SOBL-200x150-850, SmartVision Lights, Muskegon, MI). Flies were introduced into the chambers by gentle aspiration and allowed to settle for at least 2 minutes prior to testing. Behaviors were recorded from above using a Point Grey Flea3 camera recording at 30 fps using a long pass IR filter (780 nm, Midwest Optical Systems). Flies were tested during the evening peak (Zeitgeber (ZT) 9–12).

### Sliding-door assay

Detailed descriptions of the behavioral setup can be found in *Inagaki et al., 2014* and *Hoopfer et al., 2015*. Briefly, sliding-door experiments were performed in identical arenas apart from a vertical barrier that divides each chamber in half. Females were loaded into separate sides of the barrier and allowed to acclimate for 2 minutes before recording. Each trial comprised the pre-stimulation (30 seconds), stimulation (30 seconds), post-stimulation I (600 seconds), buffering (10 seconds), and post-stimulation II (600 seconds) period. The stimulation period comprises two stimulation blocks (13 and 15 seconds) separated by a 2-second inter-stimulation interval. For each stimulation block, a 655 nm LED above each chamber delivers 5 Hz pulsed light with a maximum intensity of 0.62 μW/mm$^2$. Barriers were manually removed during the buffering period, meaning that social interaction is only allowed 10 minutes after neural stimulation. Interaction during the buffering period was not tracked due to the movement of the barrier. See 'Behavioral analysis' for quantification of aggressive behavior.

### Mixed-pair assay

To distinguish target versus tester in each experimental pair, one wing of the group housed Canton-S target is cut shorter 1 day before the experiment. Flies interacted freely in the chamber during the entire trial. Each trial comprised the pre-stimulation (30 seconds), stimulation (30 seconds), and post-stimulation (120 seconds) period. The stimulation condition is the same as the sliding-door assay. See 'Behavioral analysis' for quantification of aggressive behavior.

### Density experiment

15 or 40 group-housed mated female flies were introduced via gentle aspiration into a 127 mm diameter arena as described in *Schretter et al., 2020*. Flies were tested during the morning peak (ZT 0–3) to replicate the conditions from *Schretter et al., 2020*. The arena received was performed under white light illumination from above. Flies were acclimatized to the arena for 30 seconds prior to delivery of a single constant stimulus (1 mW/cm$^2$) from below with 660 nm LEDs lasting 30 seconds. Videos in this setup were recorded from above using a Point Grey Flea3 camera with an 800 nm long pass filter (B and W filter; Schneider Optics, Hauppauge, NY) at 30 fps.

## Behavioral analysis

Flies were tracked using the Caltech FlyTracker (http://www.vision.caltech.edu/Tools/FlyTracker/) followed by automated classification of behavior with a JAABA classifier for head butting and shoving behaviors (see *Table 1*; 80% [true positive] and 83.3% [true negative] framewise performance). Due to the camera resolution, head butting could not be distinguished from shoving and an instance of head butting and shoving was defined as when a fly moved toward the other, thrusting its head or forelimbs toward the other fly. All behavioral bouts were manually curated using the Caltech FlyTracker Visualizer to eliminate false positives and false negatives. Calculations of the fraction of time spent performing a behavior were made using the score files and averaging over the period indicated. To separate out

**Table 1.** Head butting and shoving JAABA classifier.

| Classifier | True positive | True negative | False positive | False negative |
|---|---|---|---|---|
| Head butting + shoving | 80% (204) | 83.3% (1957) | 20% (51) | 16.7% (392) |

the wild-type (Canton-S) target from the genotype of interest, mixed-pair experiments were manually corrected for tracking errors using the FixErrors MATLAB GUI (https://ctrax.sourceforge.net/fixerrors.html). For density experiments, the same classifier used in *Schretter et al., 2020* was employed.

## In vivo calcium imaging

Experiments were performed as detailed in *Chiu et al., 2021*. Briefly, 6-day-old experimental flies were anesthetized on ice and head-fixed with the UV glue in their normal standing posture. The top of the fly head was immersed in fly saline, and a piece of cuticle (350 μm by 350 μm) was removed from the posterior side of the head capsule to create an imaging window. After surgery, the experimental fly was placed under a 0.8 numerical aperture (NA) ×40 objective (LUMPLFLN40XW, Olympus) and habituated for at least 5 minutes. The optical setup for two-photon imaging with optogenetic activation was described in *Inagaki et al., 2014*. Two stimulation paradigms were used: 13- and 15-second 5 Hz stimulation with 2-second inter-stimulation interval (same as the behavioral assays) or 5-minute 50 Hz stimulation (for replicating the stimulation used in *Deutsch et al., 2020*). To calculate the decay time constant (Tau, $\tau$), we fitted the GCaMP signals (a 15-second window that immediately follows the stimulation) with a one-term exponential model (MATLAB toolbox, function 'fit'). Tau is calculated as $-1/b$.

## Connectomic analysis

The primary data used for our analyses are described in *Scheffer et al., 2020*. Hemibrain data was queried using NeuPrint and v1.2.1 of the connectome (neuprint.janelia.org). Cytoscape (cytoscape.org) was used to produce the node layout of connectivity diagrams of connections between neurons, which were then edited in Adobe Illustrator. Thresholds were used to limit the number of neurons in the figures to those connections with the most synapses. For *Figure 1—figure supplement 2*, a threshold of 25 synapses between types was used, except for the connections from pC1d and pC1e in which all synapses were shown. In all cases, a threshold of three synapses was applied to connections between individual cells. Higher specific thresholds, when applied, are specified in each figure legend. A complete list of synaptic connections can be found in NeuPrint.

## Statistics and quantification

No statistical methods were used to predetermine sample size. Sample size was based on previous literature in the field and experimenters were not blinded in most conditions as almost all data acquisition and analysis were automated. Biological replicates completed at separate times using different parental crosses were performed for each of the behavioral experiments. For mixed-pair and sliding-door experiments, three biological repeats were performed, and the data was combined. For density experiments, data is representative of two independent biological repeats, only one of which is shown. For figures in which the behavioral data over the course of a trial is shown, gray shading indicates the stimulus period, the mean is represented as a solid line, and shaded error bars represent variation between experiments. For raster plots, each red line represents a bout in which either individual in the arena displayed head butting or shoving behavior. In the mixed-pair experiments, data from the wild-type target was not included.

For each experiment, the experimental and control flies were collected, treated, and tested at the same time. A Kruskal–Wallis test and Dunn's post hoc test were used for statistical analysis for behavioral experiments. For *Figure 4*, *Figure 4—figure supplement 1*, a paired *t*- test was used for statistical analysis. All statistical analyses were performed using Prism software (GraphPad, version 9). p values are indicated as follows: ****$p < 0.0001$; ***$p < 0.001$; **$p < 0.01$; and *$p < 0.05$. See *Supplementary file 2* for the exact p values for each figure.

Boxplots show median and interquartile range (IQR). Lower and upper whiskers represent 1.5×IQR of the lower and upper quartiles, respectively; boxes indicate lower quartile, median, and upper quartile, from bottom to top. When all points are shown, whiskers represent range and boxes indicate lower quartile, median, and upper quartile, from bottom to top. Shaded error bars on graphs are presented as mean ± s.e.m.

## Acknowledgements

We thank M Dreher (Dreher Design Studio) for help with connectomics figures; A Sanchez (Caltech) for fly maintenance; C Chiu, G Mancuso, L Chavarria, X Da (Caltech), and A Howard (Janelia) for laboratory management and administrative assistance; and Dr D Bushey (Janelia) for help with additional calcium imaging experiments. We also thank Drs D Deutsch, D Galili, U Heberlein, M Murthy, and A Otopalik for their helpful feedback on the manuscript as well as the broader Janelia community for their suggestions throughout this work.

## Additional information

### Funding

| Funder | Grant reference number | Author |
| --- | --- | --- |
| Howard Hughes Medical Institute | | Alice A Robie<br>Kristin Branson<br>Tanvi Vippa<br>Samantha Epstein<br>Gerald M Rubin<br>David J Anderson<br>Catherine E Schretter |
| National Institutes of Health | R37DA031389 | Hui Chiu<br>David J Anderson |

The funders had no role in study design, data collection and interpretation, or the decision to submit the work for publication.

### Author contributions

Hui Chiu, Conceptualization, Data curation, Software, Formal analysis, Investigation, Visualization, Methodology, Writing – original draft, Writing – review and editing; Alice A Robie, Software, Writing – review and editing; Kristin Branson, Software; Tanvi Vippa, Samantha Epstein, Formal analysis, Validation, Writing – review and editing; Gerald M Rubin, David J Anderson, Conceptualization, Funding acquisition, Writing – review and editing; Catherine E Schretter, Conceptualization, Data curation, Software, Formal analysis, Supervision, Validation, Investigation, Visualization, Methodology, Writing – original draft, Project administration, Writing – review and editing

### Author ORCIDs

Hui Chiu ⓘ https://orcid.org/0000-0002-1820-8411
Gerald M Rubin ⓘ https://orcid.org/0000-0001-8762-8703
David J Anderson ⓘ https://orcid.org/0000-0001-6175-3872
Catherine E Schretter ⓘ https://orcid.org/0000-0002-3957-6838

Reviewer #1 (Public Review): https://doi.org/10.7554/eLife.88598.3.sa1
Reviewer #2 (Public Review): https://doi.org/10.7554/eLife.88598.3.sa2
Author response https://doi.org/10.7554/eLife.88598.3.sa3

## Additional files

### Supplementary files
MDAR checklist

Supplementary file 1. Genotype of flies used in each experiment.

Supplementary file 2. Sample size and statistics for figures.

### Data availability
Data is included in the paper and the supplementary files. Source data have been provided for *Figures 2 and 4*, *Figure 2—figure supplements 1 and 2*, *Figure 3—figure supplement 1*, *Figure 4—figure*

*supplement 1*. Source code was adapted from *Chiu et al., 2021* and is available in GitHub (copy archived at *Chiu and Schretter, 2025*).

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
