## [Editor Report · eLife assessment]

This study by Chiu and colleagues is a **valuable** contribution to the study of the circuitry of aggressive behaviors and of mechanisms that generate persistent behavioral states. The authors find that activation of two interconnected sets of neurons results in an increase in female aggression. The data ruling out recurrent connectivity between these clusters underlying this persistent state are **convincing**.

---

## [Referee Report · Reviewer #1 (Public Review)]

Establishing direct links between the neuronal connectivity information of connectomics datasets with circuit physiology and behavior and exciting current research area in neurobiology. Until recently, studies of aggression in *Drosophila* had been conducted largely in males, and many of the neurons involved in this behavior are male-specific clusters. Since the currently available fly brain connectomes come from female brains, their applicability for the study of the circuitry underlying aggressive behavior is very limited.

The authors have previously used the Janelia hemibrain connectome paired with behavior analysis to show that activating either the aIPg or pC1d cell types can induce short term aggression in females, while activation of other PC1 clusters (a-c and e) does not. Here they expand on those findings, showing that optogenetic stimulation of aIPg neurons was sufficient to promote an aggressive internal state lasting at least 10 minutes following a 30 second activation. In addition, authors show that while stimulation of PC1d alone is not sufficient to induce this persistent aggressive state, simultaneous activation of PC1d + PC1e is, suggesting a synergistic effect. Connectomics analysis performed in the authors' previous study had shown that PC1d and aIPg are interconnected. However, silencing pC1d neuronal activity did not reduce aIPg-evoked persistent aggression, indicating that the aggressive state did not depend on pC1d-aIPg recurrent connectivity.

The conclusions are well supported by the data, and the results presented in this manuscript represent an important contribution to our understanding of the neuronal circuitry underlying female aggression.

---

## [Referee Report · Reviewer #2 (Public Review)]

The mechanisms that mediate female aggression remain poorly understood. Chiu, Schretter, and colleagues, employed circuit dissection techniques to tease apart the specific roles of particular doublesex and fruitless expressing neurons in the fly *Drosophila* in generating a persistent aggressive state. They find that activating the fruitless positive alPg neurons, generated an aggressive state that persisted for >10min after the stimulation ended. Similarly, activating the doublesex positive pC1de neurons also generated a peristent state. Activating pC1d or pC1e individually did not induce a persistent state. Interestingly, while neural activation of alPGs and pC1d+e neurons induced a persistent behavioural states it did not induce persistent activity in the neurons being activated.

The authors have revised the manuscript in accordance with comments of the reviewers. The conclusions of this paper are by and large well supported by the data. These data will be a useful addition to the literature on the circuit basis of female aggression, and open up intriguing avenues for further studies to explore.

---

## [Author Response]

The following is the authors’ response to the original reviews.

We are grateful for the comments and suggestions from the reviewers and have followed the recommendation in producing our revised manuscript. We have modified the text and performed additional statistical analysis as detailed below, which we believe has improved the overall manuscript.

**Reviewer #1 (Public Review):**
Establishing direct links between the neuronal connectivity information of connectomics datasets with circuit physiology and behavior and exciting current research area in neurobiology. Until recently, studies of aggression in *Drosophila* had been conducted largely in males, and many of the neurons involved in this behavior are male-specific clusters. Since the currently available fly brain connectomes come from female brains, their applicability for the study of the circuitry underlying aggressive behavior is very limited.The authors have previously used the Janelia hemibrain connectome paired with behavior analysis to show that activating either the aIPg or pC1d cell types can induce short-term aggression in females, while activation of other PC1 clusters (a-c and e) does not. Here they expand on those findings, showing that optogenetic stimulation of aIPg neurons was sufficient to promote an aggressive internal state lasting at least 10 minutes following a 30-second activation. In addition, the authors show that while stimulation of PC1d alone is not sufficient to induce this persistent aggressive state, simultaneous activation of PC1d + PC1e is, suggesting a synergistic effect. Connectomics analysis performed in the authors' previous study had shown that PC1d and aIPg are interconnected. However, silencing pC1d neuronal activity did not reduce aIPg-evoked persistent aggression, indicating that the aggressive state did not depend on pC1d-aIPg recurrent connectivity.The conclusions are well supported by the data, and the results presented in this manuscript represent an important contribution to our understanding of the neuronal circuitry underlying female aggression.
**Reviewer #1 (Recommendations For The Authors):**
1. Previously, the authors have shown that the activation of PC1e alone does not induce female aggression. In this study, they investigate the role of aIPg, PC1d, or PC1d+e on aggression persistence, but they do not explore the effect of activation of PC1e alone. It is possible that PC1e activation may not produce an immediate short-term effect but could lead to a gradual increase in aggression over time, potentially explaining at least in part the observed effect upon PC1d+e activation. Incorporating an examination of the long-term impact of PC1e activation on aggression could provide valuable information.

We did perform mixed pair experiments with the pC1e-SS1 line from the Schretter et al. (2020) paper and did not find any significant changes in aggression over time in this setup as well. We have now added a reference to these experiments in the revised submission in lines 135 to 136.

1. Some important controls are missing: flies with the genetic combinations employed in the activation experiments shown in Figure 2 but in the absence of activation and under the exact same conditions and for a similar observation period.

For Figure 2, we used an empty split-Gal4 driver as a genetic control for our activation paradigms. As these flies contain the same number of copies of mini-white while not labeling the targeted cell types, we believe that they provide an appropriate control for these experiments. The control information is specified in all figure legends as well.

1. The quantification shown in Fig 3- Supplementary Figure 1 shows no effect during stimulation (13 s + 15s), but based on the plots of Figure 3, there may be an effect of silencing PC1d on aIPg-induced aggression during the initial 13 second period. Those two time periods (13 s vs 15 s) could be quantified separately to determine if this is the case.

We examined the two stimulation periods separately and did not find any significant differences in either period (13s period, p = 0.2978; 15s period, p = 0.6650). We have now added this into the figure legend for Figure 3 and Figure 3 supplement 1.

1. Expression of Kir2.1 in pC1d neurons while aIPg neurons were activated did not suppress aggression after aIPg stimulation, suggesting that connections from pC1d neurons are not necessary for the persistent aggressive state promoted by aIPg. Since previously the authors have shown that TNT-mediated inhibition of aIPg reduces aggression, the reciprocal experiment would be informative: determining if stimulation of PC1d+e no longer produces persistent aggression when aIPg neurons are silenced.

In this manuscript, we were primarily testing if the connections from aIPg to pC1d were necessary for the persistent aggressive state induced by aIPg activation. Therefore, we believe the suggested experiment is beyond the scope of the current manuscript.

1. How many times was each experiment repeated? This is important information and should be in the methods section for each type of experiment or in each figure legend.

We have now added this information in the appropriate figure legends.

1. Determining the effect on persistent aggression of silencing sNPF (for example via RNAi or Crispr-Cas9 mediated mutagenesis) in aIPG neurons would be an important addition to the manuscript. If peptidergic signaling is underlying the persistence phenotype of aIPg neurons, that would explain why the recurrent connectivity found between those cells and the PC1 cluster does not play a role.

We agree with the reviewer that this would be a logical next step in extending this work.

**Reviewer #2 (Public Review):**
The mechanisms that mediate female aggression remain poorly understood. Chiu, Schretter, and colleagues, employed circuit dissection techniques to tease apart the specific roles of particular doublesex and fruitless expressing neurons in the fly *Drosophila* in generating a persistent aggressive state. They find that activating the fruitless positive alPg neurons, generated an aggressive state that persisted for >10min after the stimulation ended. Similarly, activating the doublesex positive pC1de neurons also generated a persistent state. Activating pC1d or pC1e individually did not induce a persistent state. Interestingly, while neural activation of alPGs and pC1d+e neurons induced persistent behavioural states it did not induce persistent activity in the neurons being activated.The conclusions of this paper are well supported by the data, there were only a few points where clarification might help:1. Figure 3 is a little confusing. This is a circuit behavioural epistasis experiment where the authors activate alPg with CsChrimson while inhibiting pC1d with Kir2.1. In Fig. 2 flies were separated for 10 min following stimulation which allowed for identification of a persistent state. However, in Fig 3 it appears as if flies were allowed to freely interact during and immediately post-stimulation. It is unclear why flies were not separated as in Fig. 2, which makes it difficult to compare the two results. Some discussion of this point would help. Also, from the rasters it appears as if inhibition of pC1d reduced aggression induced by alPg during the stimulation period. Is this true?

We thank the reviewer for pointing out the need for clarification and we have modified the legend in Figure 3 to address the points raised. The flies were allowed to freely interact during the experiments shown in Figure 3 and we have added this information to the figure legend. To obtain a high level of aggressive behavior that would make it easier to observe a suppression of aggression, the epistasis experiments were performed with freely moving same-genotype pairs. The level of aggression triggered by the generation 1 LexA line labeling aIPg was lower than that observed when using with the aIPg-SS GAL4 line. The experiment was performed as in Schretter et al. (2020) where we found that aIPg activation induced persistent fighting in same genotype pairs. We have added a brief explanation in lines 152 to 155.

Inhibition of pC1d does not significantly reduce the overall aggression induced by aIPg stimulation in the 13s + 15s period. We also examined the differences within the two stimulation periods and did not find any significant differences (13s period, p = 0.2978; 15s period, p = 0.6650). We have now added this information to the figure legends for Figure 3 and Figure 3 supplement 1.

1. pC1e neurons also have recurrent connectivity with alPg neurons. It might help to also discuss the potential role of this arm of the microcircuit.

We thank the review for this suggestion. The number of synapses that aIPg sends back to pC1e is a very low proportion of its total output (0.177%). However, based on the experiments that we have performed, we cannot rule out that this microcircuit might contribute to maintaining persistence. We have added this point into the discussion in lines 210 to 211.

**Reviewer #2 (Recommendations For The Authors):**
1. Line 129-130: A citation for group-housed flies showing lower aggression would be helpful.

We have now added in the reference to Chiu et al. (2021), as they showed this effect for females, in line 130.

1. Figure 2 - figure supplement 1: In the legend, change "when pC1d neurons were stimulation" to "when pC1d neurons were stimulated".

We thank the reviewer for finding this error and have now corrected this.

**Reviewer #3 (Public Review):**
Two studies published in 2020 independently identified the alPg, pC1d, and pC1e neurons to be involved in initiating and maintaining a state of aggression in female *Drosophila*. Both studies combined behavioural analyses, optogenitic manipulation of neurons, and connectomics. One of these studies proposed that the extensive interconnections seen between the alPg and pC1d+e neurons might represent a recurrent motif known to support persistent behvioural states in other systems. In this manuscript, the authors test this idea and report that their data do not support it. Specifically, they report that alPg or pC1d+e (but not pC1d alone) can initiate a persistent state of aggression. But they find that the persistent aggressive state is maintained even when the pC1d neurons are inactivated. Finally, they show that neither of these neurons themselves sustains neuronal activity upon stimulation, nor do either of them induce a persistent activity in the other. Together, their data suggest that the recurrent connection between alPg and pC1d is not what supports the persistent state. The data underlying these claims are convincing. A possibility to explore before ruling out recurrent motifs (at this circuit level) in maintaining aggression is that the connections between alPg and pC1e can compensate for the loss of pC1e. Overall, the study is important and will be of interest to those who study the circuit basis of persistent behavioural states, but also to neuroscientists in general.
**Reviewer #3 (Recommendations For The Authors):**
I enjoyed reading this manuscript for its clarity in writing and data presentation.I would like the authors to comment on the possibility that pC1e can compensate for the loss of pC1d. It is possible that if they silence both pC1d+e in the context of alPg activation, the persistent aggression is lost?

We agree with the reviewer that this is an intriguing hypothesis. In order to examine if pC1e does compensate for pC1d, we would need to also activate pC1e while inhibiting pC1d. However, such an experiment is not currently possible as we do not have a LexA line that specifically labels either pC1d or pC1e alone.

For the pC1d+e silencing experiments, we were primarily testing to see if the most prominent recurrent connection, which is between pC1d and aIPg, was responsible for the behavioral persistence. We agree with the reviewer that this would be a logical follow up experiment to be performed in the future.

Have the authors looked for activity in the pC1e neuron upon simulation of alPg? (Deutsch et al 2020 observed many regions in the brain that maintained sustained activity upon pC1d+e stimulation.)

We have not examined this activity. We agree that this would be a good follow up experiment; however, we believe it is beyond the scope of the current work.

Would the more appropriate experiment in Figure 4c be the co-stimulation of pC1d+e while imaging from alPg?

For these experiments, we were testing to see if the most prominent recurrent connection, which is between pC1d and aIPg, was responsible for the behavioral persistence. We agree with the reviewer that this would be a good follow up experiment to be performed in the future.